# Poor Nutritional Status during Recovery from Acute Myocardial Infarction in Patients without an Early Nutritional Intervention Predicts a Poor Prognosis: A Single-Center Retrospective Study

**DOI:** 10.3390/nu15224748

**Published:** 2023-11-10

**Authors:** Hiroshi Abe, Tetsuro Miyazaki, Masato Tomaru, Yuka Nobushima, Tomohi Ajima, Koji Hirabayashi, Sayaki Ishiwata, Midori Kakihara, Masaaki Maki, Ryosuke Shimai, Tadao Aikawa, Hiroyuki Isogai, Dai Ozaki, Yuki Yasuda, Fuminori Odagiri, Kazuhisa Takamura, Makoto Hiki, Hiroshi Iwata, Ken Yokoyama, Takashi Tokano, Tohru Minamino

**Affiliations:** 1Department of Cardiology, Juntendo University Urayasu Hospital, 2-1-1 Tomioka, Chiba, Urayasu-shi 279-0021, Japan; hi-abe@juntendo.ac.jp (H.A.); m.tomaru.jv@juntendo.ac.jp (M.T.); y.sugawara.dr@juntendo.ac.jp (Y.N.); t.ajima.bw@juntendo.ac.jp (T.A.); k-hirabayashi@juntendo.ac.jp (K.H.); s-ishiwata@juntendo.ac.jp (S.I.); mkakiha@juntendo.ac.jp (M.K.); mmaki@juntendo.ac.jp (M.M.); r-shimai@juntendo.ac.jp (R.S.); tadao.aikawa@juntendo.ac.jp (T.A.); h-isogai@juntendo.ac.jp (H.I.); dozaki@juntendo.ac.jp (D.O.); y-yasuda@juntendo.ac.jp (Y.Y.); fodagiri@juntendo.ac.jp (F.O.); k-takamu@juntendo.ac.jp (K.T.); kyokoyam@juntendo.ac.jp (K.Y.); taka@juntendo.ac.jp (T.T.); 2Department of Cardiovascular Biology and Medicine, Juntendo University Graduate School of Medicine, 2-1-1 Hongo, Bunkyo-ku, Tokyo 113-8421, Japan; m-hiki@juntendo.ac.jp (M.H.); h-iwata@juntendo.ac.jp (H.I.); t.minamino@juntendo.ac.jp (T.M.); 3Japan Agency for Medical Research and Development-Core Research for Evolutionary Medical Science and Technology (AMED-CREST), Japan Agency for Medical Research and Development, 1-7-1 Otemachi, Chiyoda-ku, Tokyo 100-0004, Japan

**Keywords:** acute myocardial infarction, malnutrition, GNRI, nutritional status

## Abstract

Whether malnutrition during the early phase of recovery from acute myocardial infarction (AMI) could be a predictor of mortality or morbidity has not been ascertained. We examined 289 AMI patients. All-cause mortality and composite endpoints (all-cause mortality, nonfatal stroke, nonfatal acute coronary syndrome, and hospitalization for acute decompensated heart failure) during the follow-up duration (median 39 months) were evaluated. There were 108 (37.8%) malnourished patients with GNRIs of less than 98 on arrival; however, malnourished patients significantly decreased to 91 (31.4%) during the convalescence period (*p* < 0.01). The incidence rates of mortality and primary composite endpoints were significantly higher in the malnourished group than in the well-nourished group both on arrival and during the convalescence period (All *p* < 0.05). Nutrition guidance significantly improved GNRI in a group of patients who were undernourished (94.7 vs. 91.0, *p* < 0.01). Malnourished patients on admission who received nutritional guidance showed similar all-cause mortality with well-nourished patients, whereas malnourished patients without receiving nutritional guidance demonstrated significantly worse compared to the others (*p* = 0.03). The assessment of GNRI during the convalescence period is a useful risk predictor for patients with AMI. Nutritional guidance may improve the prognoses of patients with poor nutritional status.

## 1. Introduction

Acute myocardial infarction (AMI) is a major cause of death and disability worldwide [1]. On the other hand, malnutrition is highly prevalent in patients with cardiovascular disease, and undernutrition is associated with unfavorable prognosis and mortality in those patients. In fact, several objective nutritional indices such as the Geriatric Nutrition Risk Index (GNRI) [2,3,4,5], Controlling Nutritional Status (CONUT) score [6,7,8,9], and Prognostic Nutritional Index (PNI) [4,10,11,12] have been reported as useful clinical predictors of mortality or morbidity in patients with AMI. In the acute phase of AMI, however, BMI, serum albumin levels, and total lymphocyte counts, which are key components of the GNRI, CONUT, and PNI, may fluctuate due to systemic status, time elapsed since the onset of AMI and accompanying heart failure. Therefore, the nutritional status during early recovery from AMI could be a predictor of mortality or morbidity in patients with AMI instead of the nutritional status in the acute phase of AMI. Furthermore, nutritional intervention using nutritional guidance would be an effective way to improve malnutrition status. However, it is also unclear whether interventions using early nutritional guidance improve AMI’s nutritional status and prognosis.

In this study, we investigate whether nutritional status during recovery from AMI could be one of the predictors of mortality or morbidity and whether early nutritional interventions can improve malnutrition status and reduce the risk of adverse events in patients with AMI.

## 2. Materials and Methods

### 2.1. Study Design and Population

This retrospective cohort study was conducted at Juntendo University Urayasu Hospital. We enrolled 385 consecutive patients with AMI who underwent emergency coronary angiography (CAG) and percutaneous coronary interventions (PCI) at Juntendo University Urayasu Hospital from 1 September 2015 to 30 September 2022. AMI was diagnosed by the universal definition of myocardial infarction [13]. There is an acute myocardial injury with clinical evidence of acute myocardial ischemia and with detection of a rise and/or fall on troponin values with at least one value above the 99 percentile upper reference limit and at least one of the following: symptoms of myocardial ischemia, new ischemic electrocardiogram changes, development of pathological Q waves, imaging evidence of new loss of viable myocardium or new regional wall motion abnormality in a pattern consistent with an ischemic etiology, and identification of a coronary thrombus via angiography. We excluded 96 patients who experienced in-hospital death (*n* = 31), did not visit our outpatient after discharge (*n* = 16), or had no available data for calculating GNRI in the outpatient clinic during the convalescence period (which was defined as more than three weeks but less than two months from the onset of AMI, *n* = 49); thus, we finally enrolled 289 patients with complete follow-up data to analyze the long-term prognosis. The present study was approved by the medical ethics committee at Juntendo University Hospital.

Demographic data and information about coronary risk factors, medications at discharge, and co-morbidities were prospectively collected and analyzed. As previously described, BMI was calculated from body height and weight: BMI = weight (kg)/height^2^ (m). Hypertension, dyslipidemia, diabetes, family history of coronary artery disease, and smoking were considered coronary risk factors. Hypertension is defined as people whom a health care provider has noted as patients taking antihypertensive medications for purposes other than myocardial protection. Dyslipidemia was defined as low-density lipoprotein cholesterol (LDL-C) ≥ 140 mg/dL, high-density lipoprotein cholesterol (HDL-C) ≤ 40 mg/dL, triglycerides ≥ 150 mg, or current treatment with statins and/or lipid-lowering agents [14]. Diabetes mellitus was either hemoglobin A1c (HbA1c) ≥ 6.5% or medication with insulin or oral hypoglycemic drugs [15]. A family history of cardiovascular disease was defined as a family member within the second degree of consanguinity with a physician’s diagnosis of acute myocardial infarction or angina pectoris. A smoker was described as a person who was smoking at the time of arrival in the emergency department or had stopped smoking within ten years before AMI. Chronic kidney disease (CKD) was defined as a previous history of being diagnosed with decreased kidney function as shown by a glomerular filtration rate of less than 60 mL/min per 1.72 m^2^ using the Modification of Diet in Renal Disease equation modified with a Japanese coefficient using baseline serum creatinine [16], the presence of markers of kidney damage, or both for at least three months. The estimated glomerular filtration rate was calculated based on the Japanese equation that uses serum creatinine level, age, and gender as follows: estimated glomerular filtration rate (mL/min/1.73 m^2^) = 194 × creatinine^−1.094^ × age^−0.287^ (female × 0.739). History of heart failure was defined as a diagnosis of heart failure which needed hospitalization for treatment. Atrial fibrillation was described as having been noted in the past at a medical facility or health check-up, excluding those first detected at the onset of AMI. Old myocardial infarction was defined as a previous diagnosis of myocardial infarction for which a catheterization was performed. Old cerebral infarction was described as a previous neurological abnormality and confirmed by imaging. ST-elevated myocardial infarction was defined with electrocardiographic manifestations of new ST-elevation at the J-point in 2 contiguous leads with the cut-point: ≥1 mm in all leads other than leads V_2_–V_3_ where the following cut-points apply: ≥2 mm in men ≥40 years; ≥2.5 mm in men <40 years; or ≥1.5 mm in women regardless of age [13]. The cardiologist assessed left ventricular ejection fraction (LVEF) using echocardiography upon arrival in the emergency department.

### 2.2. Primary Endpoint

Both all-cause mortality and cardiovascular mortality were evaluated during the follow-up period. Cardiovascular mortality was defined as death from coronary artery disease, heart failure, cardiogenic shock, or sudden death. Composite endpoints were also evaluated, including all-cause mortality, nonfatal stroke, nonfatal acute coronary syndrome, and hospitalization for acute decompensated heart failure. Nonfatal stroke was defined as diagnosed by abnormal neurological findings and imaging studies. Nonfatal acute coronary syndrome is characterized as symptomatic and estimated by CAG with severe stenosis or occlusion of the coronary artery. The cardiologist diagnosed acute decompensated heart failure. Mortality data and cardiovascular event data were collected from patients’ medical records.

### 2.3. Evaluations of Nutrition Indices

We evaluated the GNRI as a nutrition index during the acute and convalescence phases. The GNRI was calculated from serum albumin, body weight, and height, as previously described: GNRI = 14.89 × serum albumin (g/dL) + 41.7 × (actual body weight/ideal body weight). The actual body weight/ideal body weight was set to 1 when the body weight of the patient exceeded the ideal body weight. The ideal body weight in the present study was calculated using a body mass index (BMI) of 22 kg/m^2^ because of its validity [17] instead of the value calculated using the Lorentz formula in the original GNRI equation [18]. The height and body weight were recorded when the patient was discharged. Patients with GNRIs of less than 98 were assigned to the malnourished group, while the other patients were assigned to the well-nourished group. Blood samples were collected in the emergency department on arrival and in the outpatient department during the convalescence period.

### 2.4. Nutritional Intervention

Nationally licensed nutritionists provided nutritional guidance. The nutritional guidance used in our study was also based on the definition of adequate caloric intake of 25–30 kcal per ideal body weight (kg). Under an appropriate total energy intake, a fat energy ratio of 20–25% and a carbohydrate energy ratio of 50–60% are basically recommended, with protein and lipid restrictions depending on the disease state [19]. For patients with CKD, restricting protein (0.6–0.8 g/kg body weight per day) is recommended to reduce the risk for end-stage kidney disease [20]. For patients with dyslipidemia, the daily cholesterol intake was advised to be <200 mg [21]. In accordance with the guidelines established by the Japanese Circulation Society, the prompt initiation of meal intake is strongly recommended in AMI patients [22]. The start of enteral nutrition in patients under intubation management was defined as administering tube feedings of 1 kcal/mL or more. It was defined as initiating a general diet for patients not under intubation management. It was difficult to accurately date the time from arrival at the hospital to the start of meals on an hourly basis because the meal start differed from ward to ward. Therefore, the number of days after arrival at the hospital was used to evaluate the data.

### 2.5. Statistical Analysis

In previous reports [2,3], the mean long-term (3–5 years) mortality rates of AMI patients were approximately 12% and 25–60% in well-nourished and malnourished patients on admission, respectively. In the present study, we hypothesized that malnourished patients during the convalescence period have a higher mortality rate than patients with a good nutritional status (25% vs. 12%). According to our statistical power analysis, a minimum sample size of 278 patients was required to detect a substantial relative risk reduction with 80% statistical power and a two-sided α of 5%.

Quantitative data are presented as the mean ± standard deviation or the median and interquartile range (IQR). Categorical variables are presented as frequencies and percentages. The χ^2^ test or Fisher’s exact test was used for categorical variables to compare the baseline characteristics between the two groups. In contrast, the Student’s *t*-test or the Mann–Whitney U test was used for continuous variables. Unadjusted cumulative event rates were estimated using Kaplan–Meier curves and compared across groups. The receiver operating characteristic (ROC) curve analysis was carried out based on GNRI. The multivariate Cox regression modeling was attempted using factors associated with long-term mortality, and variables with *p*-values of less than 0.05 was entered into the Cox regression model. All statistical analyses were performed using EZR (Saitama Medical Center, Jichi Medical University, Saitama, Japan), which is a graphical user interface for R (The R Foundation for Statistical Computing, Vienna, Austria). More precisely, it is a modified version of R commander designed to add statistical functions frequently used in biostatistics [23]. Long-rank *p*-values are two-sided. A *p*-value of less than 0.05 was considered statistically significant.

## 3. Results

### 3.1. Baseline and Procedural Characteristics

Among 289 AMI cases, there were 108 (37.8%) patients estimated to be malnourished patients with GNRIs of less than 98 on arrival; however, the number of malnourished patients significantly decreased to 91 (31.4%) during the convalescence period (*p* < 0.01). The median GNRI on arrival of these study participants was 99.8 (IQR: 91.8–107.8), and this value also significantly increased to 100.3 (IQR 96.8–105.7, *p* < 0.01) during the convalescence period.

We divided the study population into well-nourished (*n* = 198, 68.5%) and malnourished (*n* = 91, 31.4%) groups based on the GNRI in the convalescence period. The characteristics of our study participants are shown in Table 1. Patients in the malnourished group during the convalescence period were significantly older, and there were more females, never-smokers, people with chronic kidney disease, and also a lower proportion of people with dyslipidemia. During the convalescence period, patients in the malnourished group also had lower BMI, longer hospitalization durations, and delayed start of oral food intake, including enteral feeding. The proportion of ST-elevated myocardial infarction and LVEF levels did not show significant differences between the two groups. The laboratory data on admission with the malnourished group showed significantly lower hemoglobin (Hb) and albumin levels, higher B-type natriuretic peptide (BNP), higher potassium (K), higher *C*-reactive protein (CRP), and higher glucose levels. The patients in the malnourished group had lower total cholesterol, LDL-C, and triglyceride (TG), while HDL-C and HbA1c levels did not differ between the two groups. Maximum creatine kinase levels, indicating damage to the myocardium, were also equivalent among well-nourished and malnourished patients. The medication at discharge was significantly fewer antiplatelet drugs (aspirin, P2Y12 inhibitor) in the malnourished group. On the other hand, the patients in the malnourished group had received more anticoagulation therapy. Angiotensin-converting enzyme inhibitors (ACE-Inhibitor) and angiotensin II receptor blocker (ARB) were significantly lower in the malnourished group. This may be due to the large number of CKD patients in the malnutrition patient group, which was considered to be at high risk for CKD worsening; thus, the introduction of the system was abandoned.

### 3.2. Clinical Outcomes

The median follow-up duration was 39 months (IQR: 13.5–64.5 months). In total, 17 all-cause deaths (5.9%) were identified during follow-up, including only four cases of cardiovascular mortality (1.4%). In addition, 10 cases of nonfatal stroke (3.5%), 45 cases of nonfatal acute coronary syndrome (15.6%), and 15 cases of hospitalization for acute decompensated heart failure were identified.

All-cause mortality and primary composite endpoints among patients stratified by nutritional status using GNRI on arrival and during the convalescence period are presented in Figure 1. Kaplan–Meier curves showed that the incidence rates of death and primary composite endpoints were significantly higher in the malnourished group than in the well-nourished group both on arrival and in the convalescence period. Focusing on each event, only the incidence of hospitalization for acute decompensated heart failure showed a significant difference between the malnourished and well-nourished groups on arrival (9.3% vs. 2.8%, *p* = 0.03). On the other hand, the incidence of nonfatal stroke and hospitalization for acute decompensated heart failure showed significant differences between the malnourished and well-nourished groups during the convalescence period (9.9% vs. 0.5%, *p* < 0.01; 9.9% vs. 3.0%, *p* = 0.02, respectively). Both the incidences of cardiovascular death and nonfatal acute coronary syndrome did not show any differences between well-nourished and malnourished groups on admission and in the convalescence period. The ROC curves for all-cause mortality and the primary composite endpoint did not differ significantly between GNRI on arrival and the convalescence period (AUC = 0.68 vs. 0.72, *p* = 0.378; AUC = 0.73 vs. 0.76, *p* = 0.258, respectively). These results suggested that the predictive value of GNRI during the convalescence period was equivalent to GNRI estimated on admission.

The multivariate Cox regression model used all factors which had statistically significant associations with long-term mortality (age, gender, Hb, Cre, BNP, and Cl) and primary composite endpoints (age, gender, Hb, Cre, BNP, K, HDL-C, and LDL-C). Low GNRI on arrival and during convalescence were each significant predictors of mortality. Still, GNRI on arrival was not a significant factor in the multivariate Cox regression, while GNRI during convalescence was. Although GNRI on arrival was not a significant factor in primary composite endpoints, Low GNRI during the convalescence period was also a significant factor in multivariate Cox regression for primary composite endpoints (Table 2). Thus, GNRI in the convalescence period, but not GNRI on arrival, was an independent predictor of future cardiovascular events.

Nutrition guidance by nationally licensed nutritionists was provided in 162 cases (56.1%) until the evaluation of nutrition indices during the convalescence period. Nutritional guidance in our study was based on the definition of adequate caloric intake (which was defined as 25–30 Kcal per ideal body weight (Kg)), with protein and lipid restriction being implemented, depending on the disease state. No significant improvement in GNRI by early nutritional guidance was observed in well-nourished patients on arrival; however, nutrition guidance significantly improved GNRI during the convalescence period in a group of undernourished patients on arrival (Figure 2).

Patients with poor nutritional status who received nutritional guidance on admission showed similar all-cause mortality with those who had a good nutritional status, whereas patients with poor nutritional status who did not receive nutritional guidance demonstrated significantly worse all-cause mortality compared to the others. In the primary composite endpoint, the effects of nutritional guidance during early recovery from AMI were limited (Figure 3A,B).

## 4. Discussion

### 4.1. Clinical Implication

Previous studies have reported that the GNRI upon hospital arrival is associated with the prognosis of AMI [2,3,4,5]; however, there have been no reports on nutritional management or nutritional indices after the onset of AMI. In the present study, we defined the convalescence period as more than three weeks from AMI onset. We found that the GNRI during the convalescence period was associated with all-cause mortality and composite primary endpoints. The multivariate Cox regression analysis revealed the superiority of GNRI during the convalescence period as compared to that on admission. Furthermore, the present study demonstrated that nutritional guidance during the early convalescence phase improved the nutritional status and all-cause mortality on hospital arrival in undernourished patients. These results suggested that nutritional assessments and early nutritional guidance during the convalescence period after AMI should be performed as part of routine care for AMI patients in future clinical settings.

### 4.2. Malnutrition on Convalescence Period in AMI Patients

Although several reports (including ours) demonstrated the usefulness of GNRI on admission as a predictor in various cardiovascular diseases [24,25,26,27], whether or not GNRI during the acute phase of cardiovascular disease is an appropriate long-term predictor even after weaning from the acute phase has been controversial. Actually, both BMI and serum albumin levels (which are components of the GNRI) may fluctuate due to systemic status, time elapsed since the onset of AMI, and accompanying heart failure. Furthermore, measuring the body mass index accurately in the emergency phase of acute coronary syndrome may be difficult. In this study, we found that the GNRI during the convalescence period was significantly improved compared with that on admission, suggesting that the GNRI on admission may not reflect the nutritional status after recovery from the acute phase of AMI. GNRI during the convalescence period may be a more appropriate long-term prognostic marker because it is evaluated in more stable conditions after withdrawal from the acute condition. Although the ROC curves of GNRI both for all-cause mortality and the primary composite endpoint did not differ significantly between the moment of admission and the convalescence period, our results suggested that the GNRI during the convalescence period after AMI is a useful long-term predictor, as well as the GNRI estimated during the acute phase. Although the convalescence period was defined as more than three weeks but less than two months from the onset of AMI in this study, the optimal time to assess nutritional status from the onset of AMI should be considered in future studies.

### 4.3. Enteral Nutrition

In this study, the well-nourished group during the convalescence period was associated with a significantly earlier diet start after the onset of AMI, including ventilator-managed patients who commenced enteral nutrition earlier. The start of enteral nutrition in patients under intubation management was defined as administering tube feedings of 1 kcal/mL or more. In heart failure, it is recommended that enteral nutrition be started within 48 h of ICU admission [28]. In this study, the proportion of patients who started enteral nutrition by the second day of illness was significantly higher in the group with good GNRI during the convalescence period. Taken together, clinicians should start diets earlier after AMI to maintain a good nutritional status.

### 4.4. Nutritional Guidance

Nutritional guidance for patients with acute myocardial infarction generally focused on improving lifestyle-related diseases such as hypertension, diabetes mellitus, and dyslipidemia from the secondary prevention perspective [29] but not improving malnutrition. The appropriate diet for weight loss (respecting the recommended calorie intake) is usually recommended for lifestyle-related diseases. Nutritional guidance in our study was also based on the definition of adequate calorie intake as 25–30 Kcal per ideal body weight (Kg), with protein and lipid restriction depending on the disease state and performed by nationally licensed nutritionists. In this study, appropriate caloric intake guidance for undernourished patients with AMI rapidly improved the GNRI, which comprises BMI and albumin levels, and prevented all-cause mortality. Previous reports have demonstrated the usefulness of nutrition support in reducing the risk of mortality and major cardiovascular events among hospitalized patients with chronic heart failure [30]. Our study showed that early intervention using appropriate nutritional guidance may improve the prognosis of patients with AMI due to slight but significant improvements in the nutritional status. If more effective methods of nutritional guidance for malnourished patients suffering from AMI are established, there is potential for further improvement in the mortality of AMI patients.

### 4.5. New Directions for Future Research

Our single-center retrospective observational study demonstrated that a low GNRI during the convalescence period was useful for predicting poor prognosis in patients with AMI. Further multi-center prospective trials, however, will be needed to reconfirm our results and to investigate the usefulness of other nutritional indices, including the CONUT score and PNI, which were previously reported as predictors of AMI on hospital arrival, and unknown potential factors linked to nutritional status. In addition, BMI and serum albumin levels may fluctuate due to the existence of heart failure. Thus, it may become essential to explore alternative indicators of malnutrition independent of body fluid volume, such as sarcopenia. Therefore, sarcopenia indicated by low muscle strength, low muscle quantity, and poor physical performance [31] should be verified as an alternative indicator of malnutrition in patients with cardiovascular disease.

Furthermore, the nutritional guidance and early start of diet after AMI were effective to improve the patients’ nutritional status and prognosis. These results warrant future prospective randomized trials to clarify the contents of the nutritional guidance, including optimal dietary calories, balance of carbohydrates, proteins, and fats, active use of nutritional supplements, timing of administration, initial caloric content, and titration methods.

### 4.6. Study Limitation

First, this was a single-center observational study, which could have inherent limitations, such as the inability to adjust for unmeasured confounders. Moreover, chart review data inevitably have some missing values, which differ in frequency according to the variables. These missing data points and underestimated frequencies may have affected the analysis results. Second, despite the careful chart review, the frequency of adverse clinical events may have been underestimated. Third, the convalescence period was defined as more than three weeks but less than two months from the onset of AMI in this study. Still, the most appropriate time to assess nutritional status and the most optimal methods with which to evaluate nutritional status remain unclear. Fourth, the optimal dietary calories, the balance of carbohydrates, proteins, and fats, the timing of administration, and the titration methods need to be clarified. Finally, it is unclear whether early nutritional intervention improves the nutritional status and the prognosis, as it is impossible to determine whether the late initiation of feeding worsens the nutritional status or whether the general condition preventing the initiation of feeding worsens the nutritional status.

## 5. Conclusions

The assessment of nutrition indices, not only in the acute phase but also during the convalescence period, is useful for risk stratification of patients with AMI. Nutritional guidance and early start of diet after AMI have the potential to improve the nutritional status and prognosis of patients, including those with poor nutritional status. Nutritional guidance and nutritional assessments during the convalescence period after AMI may become a part of routine care for patients with AMI.

## Figures and Tables

**Figure 1 nutrients-15-04748-f001:**
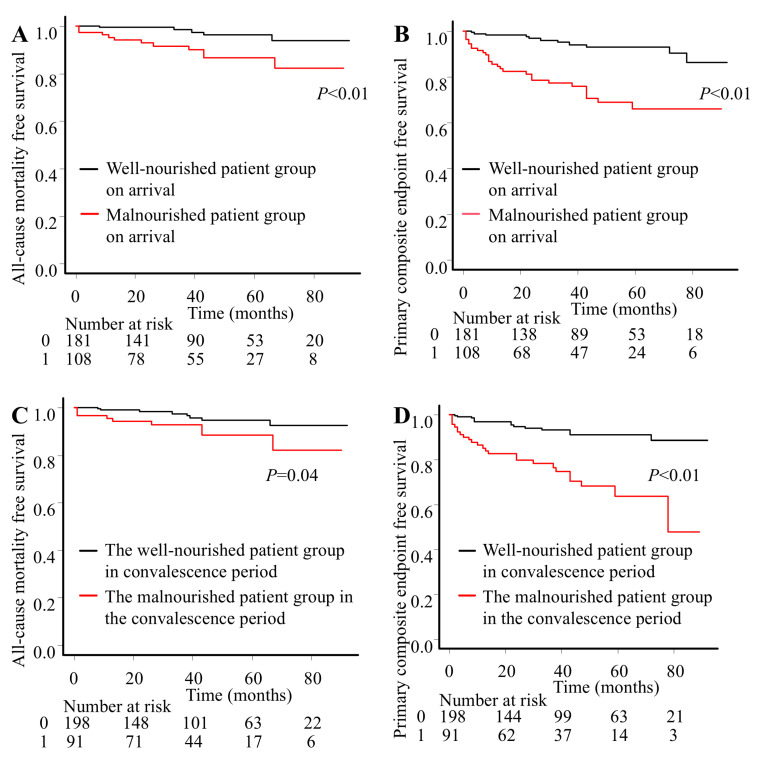
Poor nutritional status on arrival was significantly associated with all-cause mortality (**A**) and primary composite endpoints (**B**). Poor nutritional status during the convalescence period was significantly associated with all-cause mortality (**C**) and primary composite endpoints (**D**).

**Figure 2 nutrients-15-04748-f002:**
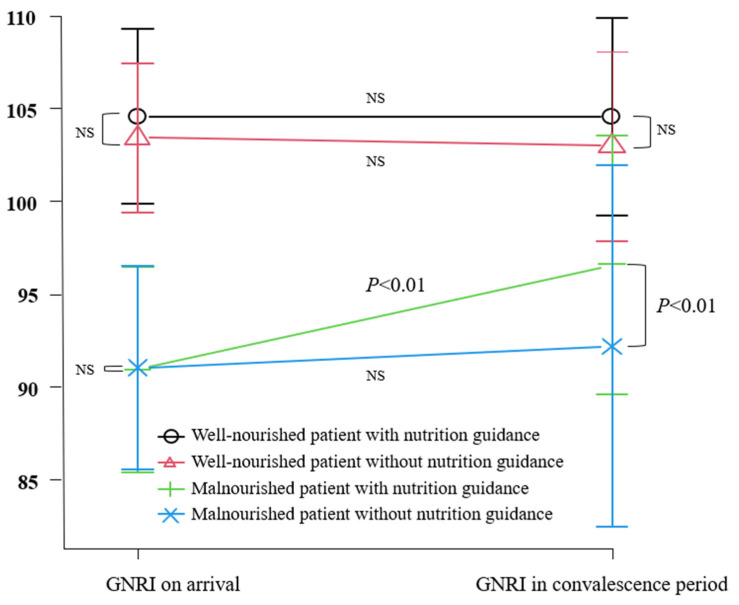
No significant improvement in the GNRI by early nutritional guidance was observed in well-nourished patients on arrival; however, nutrition guidance significantly improved GNRI during the convalescence period in a group of patients who were undernourished on arrival. NS = not significant.

**Figure 3 nutrients-15-04748-f003:**
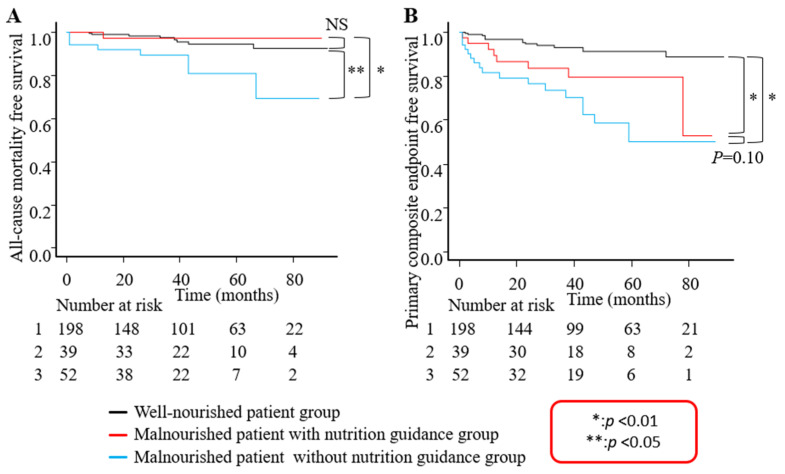
Patients with poor nutritional status who received nutritional guidance on admission showed similar all-cause mortality with those who had a good nutritional status, whereas patients with poor nutritional status who did not receive nutritional guidance demonstrated significantly worse all-cause mortality compared to the others (**A**). In the primary composite endpoint, the effects of nutritional guidance during the early phase of recovery from AMI were limited (**B**).

**Table 1 nutrients-15-04748-t001:** Baseline characteristics of the study population.

Characteristics	Well-Nourished Patient	Malnutrition Patient	*p*-Value
*n* = 289	198	91	
Age	61.50 [52.00, 72.00]	73.00 [64.50, 80.00]	<0.001
Male (*n*, %)	164 (82.8)	55 (60.4)	<0.001
BMI	24.71 [22.83, 26.98]	22.86 [20.63, 25.57]	<0.001
Hypertension (*n*, %)	130 (65.7)	68 (74.7)	0.135
Dyslipidemia (*n*, %)	166 (83.8)	60 (65.9)	0.001
Diabetes (*n*, %)	75 (37.9)	41 (45.1)	0.301
Family history of CVD (*n*, %)	41 (20.7)	14 (15.4)	0.335
Smoking (*n*, %)	138 (69.7)	51 (56.0)	0.033
CKD (*n*, %)	36 (18.2)	35 (38.5)	<0.001
History of HF (*n*, %)	2 (1.0)	3 (3.3)	0.182
Atrial fibrillation (*n*, %)	3 (1.5)	5 (5.5)	0.114
Old myocardial infarction (*n*, %)	17 (8.6)	4 (4.4)	0.233
Old cerebral infarction (*n*, %)	10 (5.1)	7 (7.7)	0.422
ST-elevated myocardial infarction (*n*, %)	181 (91.4)	78 (85.7)	0.150
Hospitalization day (*day*)	12.00 [9.00, 16.00]	20.00 [14.00, 31.50]	<0.001
LVEF on arrival *(%)*	50.00 [40.00, 55.00]	45.00 [40.00, 55.00]	0.192
Nutrition guidance (*n*, %)	123 (62.1)	39 (42.9)	0.003
Start of enteral nutrition (day)	1.74 [1.00, 2.00]	2.53 [2.00, 3.00]	<0.001
Laboratory data on arrival			
Hemoglobin (g/dL)	14.80 [13.83, 15.78]	13.50 [11.70, 15.05]	<0.001
Albumin (g/dL)	4.10 [3.80, 4.40]	3.60 [3.30, 3.90]	<0.001
Creatine kinase max (IU/L)	1712.00 [750.00, 2914.25]	1406.00 [731.50, 3448.50]	0.973
Creatinine (mg/dL)	0.81 [0.68, 1.00]	0.91 [0.70, 1.19]	0.049
eGFR (mL/min)	69.00 [57.00, 85.00]	58.00 [41.00, 77.00]	<0.001
BNP (pg/mL)	27.60 [10.30, 100.42]	63.90 [29.05, 254.60]	<0.001
Na (mEq/dL)	140.00 [138.00, 141.00]	140.00 [137.00, 142.00]	0.834
Cl (mEq/dL)	101.00 [99.00, 103.00]	101.00 [99.00, 103.00]	0.978
K (mEq/dL)	3.90 [3.60, 4.10]	4.00 [3.80, 4.40]	0.001
CRP (mg/dL)	0.30 [0.10, 0.30]	0.30 [0.30, 0.80]	<0.001
Glucose (mg/dL)	150.00 [123.75, 194.50]	161.00 [129.00, 234.50]	0.035
HbA1c (NGSP) (%)	6.10 [5.70, 6.70]	6.00 [5.70, 6.50]	0.867
T-Cho (mg/dL)	210.00 [180.50, 242.50]	192.00 [163.00, 226.75]	0.002
HDL-C (mg/dL)	45.50 [39.00, 56.00]	48.00 [40.00, 59.75]	0.299
LDL-C (mg/dL)	129.50 [104.00, 155.75]	114.00 [87.00, 139.00]	0.001
TG (mg/dL)	143.50 [96.25, 211.00]	114.50 [80.00, 165.25]	0.002
Medication at discharge			
ASA (*n*, %)	193 (97.5)	81 (89.0)	0.007
P2Y12 inhibitor (*n*, %)	194 (98.0)	83 (91.2)	0.011
OAC (*n*, %)	21 (10.6)	19 (20.9)	0.027
ACE-Inhibitor/ARB (*n*, %)	186 (93.9)	78 (85.7)	0.025
ARNI (*n*, %)	3 (1.5)	1 (1.1)	1
MRA (*n*, %)	50 (25.3)	26 (28.6)	0.567
βblocker (*n*, %)	189 (96.4)	86 (94.5)	0.528
Statin (*n*, %)	194 (98.0)	88 (96.7)	0.682
SGLT-2 inhibitor (*n*, %)	29 (14.6)	8 (8.8)	0.189
**Nutrition index**			
GNRI on arrival	102.75 [98.28, 105.73]	93.82 [89.35, 98.28]	<0.001
GNRI in convalescence period	104.24 [101.26, 107.22]	93.76 [89.08, 96.66]	<0.001

Variables are expressed as mean ± standard deviation, median (interquartile range), or *n* (%). ACE-I, angiotensin-converting enzyme inhibitor; AF, atrial fibrillation; ARB, angiotensin II receptor blocker; ARNI, Angiotensin receptor-neprilysin inhibition with LCZ696; BMI, body mass index; BNP, B-type natriuretic peptide; CKD, chronic kidney disease; Cl, chlorine; CRP, *C*-reactive protein; CVD, cardiovascular disease; eGFR, estimated glomerular filtration rate; GNRI, geriatric nutritional risk index; HF, heart failure; K, potassium; LVEF, left ventricular ejection fraction; MRA, mineral corticoid receptor antagonist; Na, natrium; OAC, oral anticoagulant therapy; SGLT-2 inhibitor; sodium glucose transporter inhibitor.

**Table 2 nutrients-15-04748-t002:** Multivariate Cox regression analysis for all-cause mortality.

	HR	95%CI	*p*-Value	Adjusted HR	95%CI	*p*-Value
Mortality						
GNRI on arrival	0.936	0.890–0.985	0.01	0.976	0.906–1.053	0.53
GNRI in convalescence period	0.924	0.891–0.958	<0.01	0.929	0.886–0.974	<0.01
Primary composite endpoints						
GNRI on arrival	0.931	0.902–0.962	<0.01	0.969	0.917–1.024	0.26
GNRI in convalescence period	0.931	0.908–0.956	<0.01	0.9401	0.902–0.979	0.01

Mortality was adjusted for age, gender, Hb, Cre, BNP, and Cl. Primary composite endpoints were adjusted for age, gender, Hb, Cre, BNP, K, HDL-C, and LDL-C.

## Data Availability

Data are contained within the article.

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
