# Peer review of "Poor Nutritional Status during Recovery from Acute Myocardial Infarction in Patients without an Early Nutritional Intervention Predicts a Poor Prognosis: A Single-Center Retrospective Study"

_nutrients, 2023, doi:10.3390/nu15224748_

Round 1

Reviewer 1 Report

Comments and Suggestions for Authors

Dear Authors,

Thank you for an interesting manuscript. Let's focus on a few suggestions:

1. Line 138 - What exactly does "protein and lipid restriction based on disease state" mean? In this case, please provide nutritional details.

2. Line 288 - enteral nutrition, not enteral nutrition

3. What is your opinion on body mass index as a scale for assessing malnutrition? Is there any other better scale? Please write some practical implications.

4. Please describe in a few words the nutritional guidelines provided and identify early dietary recommendations.

Author Response

To reviewer #1

We appreciate the valuable comments of Reviewer #1.

We revised our manuscript following the reviewer’s recommendation.

Reviewer’s comment 1: Line 138 - What exactly does "protein and lipid restriction based on disease state" mean? In this case, please provide nutritional details.

Response 1: Following the reviewer’s comment, we have added data about “protein and lipid restrictions based on the disease state” into the Methods section, which are as follows.

Line 138.

Nationally licensed nutritionists provided nutritional guidance. The nutritional guidance used in our study was also based on the definition of adequate caloric intake of 25–30 kcal per ideal body weight (kg). Under an appropriate total energy intake, a fat energy ratio of 20%–25% and a carbohydrate energy ratio of 50%–60% are basically recommended, with protein and lipid restrictions depending on the disease state [17]. For patients with CKD, restricting protein (0.6–0.8 g/kg body weight per day) is recommended to reduce the risk for end-stage kidney disease [18]. For patients with dyslipidemia, the daily cholesterol intake was advised to be <200 mg [19].

  1. Ministry of Health, Labour and Welfare of Japan. 2020. Dietary Reference Intakes for Japanese 2020. Daiichi Shuppan, Tokyo.
  2. Alp Ikizler, T.A.; Jerrilynn D. Burrowes, J.B. KDOQI Clinical Practice Guideline for Nutrition in CKD: 2020 Update. Am J Kidney Dis, 2020. 76(3 Suppl 1): p. S1-s107.
  3. Clifford Johnson, C.J.; Philip Greenland, P.G. Effects of Exercise, Dietary Cholesterol, and Dietary Fat on Blood Lipids. Archives of Internal Medicine, 1990. 150(1): p. 137.

Reviewer’s comment 2: Line 288 - enteral nutrition, not entaral nutrition

Response 2: Following the reviewer’s comment, we revised the manuscript as below.    

Line 305.

4.2. Enteral nutrition

Reviewer’s comment 3: What is your opinion on body mass index as a scale for assessing malnutrition? Is there any other better scale? Please write some practical implications.

Response 3: In patients with cardiovascular disease, BMI may fluctuate due to systemic status, time elapsed since AMI onset, and accompanying heart failure. Given this context, it becomes essential to explore alternative indicators of malnutrition independent of body fluid volume, such as sarcopenia. In 2018, the Working Group (EWGSOP2) updated its initial definition of sarcopenia to take into account scientific and clinical evidence that came during the last 10 years. The new consensus focuses on low muscle strength as a key characteristic of sarcopenia (cutoff points: grip strength < 27 kg for men and < 16 kg for women and chair stand > 15 s for five rises for both sexes), uses the detection of low muscle quantity and to confirm the sarcopenia diagnosis (cutoff points: appendicular skeletal muscle mass < 20 kg for men and < 15 kg for women), and identifies poor physical performance as indicative of severe sarcopenia (cutoff points: gait speed ≤ 0.8 m/s) [29]. Thus, instead of BMI, sarcopenia should be verified as an alternative indicator of malnutrition in patients with cardiovascular disease.

Following the reviewer’s comment, we have added relevant texts about BMI and other potential scales into the Discussion section below.

Line 339.

In addition, BMI and serum albumin levels may fluctuate due to the existence of heart failure. Thus, it may become essential to explore alternative indicators of malnutrition independent of body fluid volume, such as sarcopenia. Therefore, sarcopenia indicated by low muscle strength, low muscle quantity, and poor physical performance [29] should be verified as an alternative indicator of malnutrition in patients with cardiovascular disease.

  1. Cruz-Jentoft, A.J., et al., Sarcopenia: revised European consensus on definition and diagnosis. Age Ageing, 2019. 48(1): p. 16-31.

Reviewer’s comment 4: Please describe in a few words the nutritional guidelines provided and identify early dietary recommendations.

Response 4: Following the reviewer’s comment, we have added details about the nutritional guidelines and early dietary recommendations of our study into the Methods section, which are presented below.

Line 138.

Nationally licensed nutritionists provided nutrition guidance. Nutritional guidance in our study was also based on the definition of adequate calorie intake as 25–30 Kcal per ideal body weight (Kg). Under an appropriate total energy intake, a fat energy ratio of 20-25% and a carbohydrate energy ratio of 50-60% are basically recommended, with protein and lipid restriction depending on the disease state [17].

  1. Ministry of Health, Labour and Welfare of Japan. 2020. Dietary Reference Intakes for Japanese 2020. Daiichi Shuppan, Tokyo.

Line 145.

In accordance with the guidelines established by the Japanese Circulation Society, the prompt initiation of meal intake is strongly recommended in AMI patients [20]. The start of enteral nutrition in intubated patients was defined as administering tube feedings of ≥1 kcal/ml.

  1. Kimura, K., et al., JCS 2018 Guideline on Diagnosis and Treatment of Acute Coronary Syndrome. Circulation Journal, 2019. 83(5): p. 1085-1196.

Reviewer 2 Report

Comments and Suggestions for Authors

The manuscript aims to investigate whether nutritional status during recovery from acute myocardial infraction (AMI) can be one of the predictors of mortality or morbidity and whether early nutritional interventions can improve malnutrition status and reduce the risk of adverse events in patients with AMI. The study of very important clinical implication, well conducted and written, however some amendments are needed   

·      In the title, the design of the study should be specified i.e. prospective longitudinal study etc.

·      A clear power analysis should be reported in order to justify the sample size.

·      It is not that clear if authors included the completers only? Or some participants in the sample have dropped from the follow up, which is normal in any study. If yes, how authors dealt with that (i.e. intent-to-treat analysis?)

·      Two further subsections should be included in the Discussion section: 1) Clinical implication of the study and 2) New directions for future research on the topic.  

Author Response

To reviewer #2

Reviewer’s general comment: The manuscript aims to investigate whether nutritional status during recovery from acute myocardial infraction (AMI) can be one of the predictors of mortality or morbidity and whether early nutritional interventions can improve malnutrition status and reduce the risk of adverse events in patients with AMI. The study is very important clinical implication, well conducted and written, however some amendments are needed.

Response: We appreciate the valuable comments of Reviewer #2. We revised our manuscript following the reviewer’s recommendation.

Reviewer’s comment 1: In the title, the design of the study should be specified i.e. prospective longitudinal study etc.

Response 1: Following the reviewer’s comment, we revised the title of this article as below.

Title: Poor nutritional status during recovery from acute myocardial infarction in patients without an early nutritional intervention predicts a poor prognosis: A single-center retrospective study

Reviewer’s comment 2:  A clear power analysis should be reported in order to justify the sample size.

Response 2: Following the reviewer’s comment, we calculated the statistical power for our study design and added the results to the Methods section, which are describe below.

Line 154.

In previous reports [2, 3], the mean long-term (3–5 years) mortality rates of AMI patients were approximately 12% and 25%–60% in well-nourished and malnourished patients on admission, respectively. In the present study, we hypothesized that malnourished patients during the convalescence period have a higher mortality rate than patients with a good nutritional status (25% vs. 12%). According to our statistical power analysis, a minimum sample size of 278 patients was required to detect a substantial relative risk reduction with 80% statistical power and a two-sided a of 5%.

Reviewer’s comment 3:  It is not that clear if authors included the completers only? Or some participants in the sample have dropped from the follow up, which is normal in any study. If yes, how authors dealt with that (i.e. intent-to-treat analysis?)

Response 3: Among the 385 AMI patients, we excluded 96 patients who experienced in-hospital death, did not visit our outpatient department after discharge, or had no available data for GNRI calculation in the outpatient clinic during the convalescence period before the analysis. We finally enrolled 289 patients with complete follow-up data to analyze the long-term prognosis. Following the reviewer’s comment, we revised the manuscript to clarify the study design and methods of patient tracking, which are described below.

Line 73.

We excluded 96 patients who experienced in-hospital death (n = 31), did not visit our outpatient after discharge (n = 16), or had no available data for calculating GNRI in the outpatient clinic during the convalescence period (which was defined as more than three weeks but less than two months from the onset of AMI, n = 49); thus, we finally enrolled 289 patients with complete follow-up data to analyze the long-term prognosis. The present study was approved by the medical ethics committee at Juntendo University Hospital.

Reviewer’s comment 4:  Two further subsections should be included in the Discussion section: 1) Clinical implication of the study and 2) New directions for future research on the topic. 

Response 4: Following the reviewer’s comment, we have added two subsections into the Discussion section and deleted the redundant descriptions, which are described below.

Line 271.

4.1. Clinical implication

Previous studies have reported that the GNRI upon hospital arrival is associated with the prognosis of AMI [2-5]; however, there have been no reports on nutritional management or nutritional indices after the onset of AMI. In the present study, we defined the convalescence period as more than three weeks from AMI onset. We found that the GNRI during the convalescence period was associated with all-cause mortality and composite primary endpoints. The multivariate Cox regression analysis revealed the superiority of GNRI during the convalescence period as compared to that on admission. Furthermore, the present study demonstrated that nutritional guidance during the early convalescence phase improved the nutritional status and all-cause mortality on hospital arrival in undernourished patients. These results suggested that nutritional assessments and early nutritional guidance during the convalescence period after AMI should be performed as part of routine care for AMI patients in future clinical settings.

Line 301.

Although the convalescence period was defined as more than three weeks but less than two months from the onset of AMI in this study, the optimal time to assess nutritional status from the onset of AMI should be considered in future studies. In addition, it is necessary to consider optimal evaluation methods for nutritional status using other indices such as CONUT score and PNI, which were previously reported as predictors of AMI on arrival.

Line 311.

In this study, the proportion of patients who started enteral nutrition by the second day of illness was significantly higher in the group with good GNRI during the convalescence period. Taken together, clinicians should start diets earlier after AMI to maintain a good nutritional status. Further study will be required to clarify the optimal dietary calories, balance of carbohydrates, proteins, and fats, active use of nutritional supplements, timing of administration, initial calorie content, and titration methods.

Line 361.

Finally, it is unclear whether early nutritional intervention improves the nutritional status and the prognosis, as it is impossible to determine whether the late initiation of feeding worsens the nutritional status or whether the general condition preventing the initiation of feeding worsens the nutritional status. Future randomized trials will determine whether nutritional guidance interventions improve the prognosis of patients with AMI.

Line 333.

4.6. New directions for future research

Our single-center retrospective observational study demonstrated that a low GNRI during the convalescence period was useful for predicting poor prognosis in patients with AMI. Further multi-center prospective trials, however, will be needed to reconfirm our results and to investigate the usefulness of other nutritional indices, including the CONUT score and PNI, which were previously reported as predictors of AMI on hospital arrival, and unknown potential factors linked to nutritional status. Additionally, BMI and serum albumin levels may fluctuate due to the existence of heart failure. Thus, it may become essential to explore alternative indicators of malnutrition independent of body fluid volume, such as sarcopenia. Therefore, sarcopenia indicated by low muscle strength, low muscle quantity, and poor physical performance [29] should be verified as an alternative indicator of malnutrition in patients with cardiovascular disease.

Furthermore, the nutritional guidance and early start of diet after AMI were effective to improve the patients’ nutritional status and prognosis. These results warrant future prospective randomized trials to clarify the contents of the nutritional guidance, including optimal dietary calories, balance of carbohydrates, proteins, and fats, active use of nutritional supplements, timing of administration, initial caloric content, and titration methods.

  1. Cruz-Jentoft, A.J., et al., Sarcopenia: revised European consensus on definition and diagnosis. Age Ageing, 2019. 48(1): p. 16-31.

Round 2

Reviewer 2 Report

Comments and Suggestions for Authors

Thankful, to authors for being responsive. Still I have one single minor comment:

1. I noticed that the reference section is so poor (<30), authors are kindly ask to enrich. 

Author Response

We appreciate the comment of Reviewer #2.

Reviewer’s comment: I noticed that the reference section is so poor (<30), authors are kindly asked to enrich. 

Response: We revised our manuscript following the reviewer’s recommendation. We have added two additional references as below. Total number of references became more than 30.

Line 86.

Dyslipidemia was defined as low-density lipoprotein cholesterol (LDL-C) ≥ 140mg/dl, high-density lipoprotein cholesterol (HDL-C) ≤ 40mg/dl, triglycerides≥150mg, or current treatment with statins and/or lipid-lowering agents [14]. Diabetes mellitus was either hemoglobin A1c (HbA1c) ≥ 6.5% or medication with insulin or oral hypoglycemic drugs [15].

  1. Tamio Teramoto, T. Jun Sasaki, J.S.; Diagnostic criteria for dyslipidemia. J Atheroscler Thromb, 2013. 20(8): p. 655-60.
  2. Standards of medical care in diabetes--2010. Diabetes Care, 2010. 33 Suppl 1(Suppl 1): p. S11-61.
